# The Efficacy and Safety of High-Dose Cholecalciferol Therapy in Hemodialysis Patients

**DOI:** 10.3390/biomedicines12020377

**Published:** 2024-02-06

**Authors:** Agnieszka Tarasewicz, Michał Komorniczak, Agnieszka Zakrzewska, Bogdan Biedunkiewicz, Sylwia Małgorzewicz, Magdalena Jankowska, Katarzyna Jasiulewicz, Natalia Płonka, Małgorzata Dąbrowska, Alicja Dębska-Ślizień, Leszek Tylicki

**Affiliations:** 1Department of Nephrology, Transplantology and Internal Diseases, Medical University of Gdańsk, Smoluchowskiego 17, 80-214 Gdańsk, Poland; michal.komorniczak@gumed.edu.pl (M.K.); agnieszka.zakrzewska@gumed.edu.pl (A.Z.); bogdan.biedunkiewicz@gumed.edu.pl (B.B.); magdalena.jankowska@gumed.edu.pl (M.J.); katarzyna.jasiulewicz@gumed.edu.pl (K.J.); natalia.plonka@gumed.edu.pl (N.P.); adeb@gumed.edu.pl (A.D.-Ś.); 2Department of Clinical Nutrition, Medical University of Gdańsk, Dębinki 7, 80-211 Gdańsk, Poland; sylwia.malgorzewicz@gumed.edu.pl; 3Central Clinical Laboratory, University Clinical Center, Smoluchowskiego 17, 80-214 Gdańsk, Poland; mdabrowska@uck.gda.pl

**Keywords:** vitamin D, cholecalciferol, 25-hydroxycholecalciferol, calcidiol, calcitriol, hemodialysis, parathyroid hormone, chronic kidney disease, chronic kidney disease–mineral and bone disorder

## Abstract

Vitamin D deficiency and insufficiency are highly prevalent in CKD, affecting over 80% of hemodialysis (HD) patients and requiring therapeutic intervention. Nephrological societies suggest the administration of cholecalciferol according to the guidelines for the general population. The aim of the observational study was to evaluate the efficacy and safety of the therapy with a high dose of cholecalciferol in HD patients with 25(OH)D deficiency and insufficiency to reach the target serum 25(OH)D level > 30 ng/mL. A total of 22 patients (16 M), with an average age of 72.5 ± 13.03 years and 25(OH)D concentration of 13.05 (9.00–17.90) ng/mL, were administered cholecalciferol at a therapeutic dose of 70,000 IU/week (20,000 IU + 20,000 IU + 30,000 IU, immediately after each dialysis session). All patients achieved the target value > 30 ng/mL, with a mean time of 2.86 ± 1.87 weeks. In the first week, the target level of 25(OH)D (100%) was reached by 2 patients (9.09%), in the second week by 15 patients (68.18%), in the fourth week by 18 patients (81.18%), and in the ninth week by all 22 patients (100%). A significant increase in 1,25(OH)_2_D levels was observed during the study. However, only 2 patients (9.09%) achieved a concentration of 1,25(OH)_2_D above 25 ng/mL—the lower limit of the reference range. The intact PTH concentrations remained unchanged during the observation period. No episodes of hypercalcemia were detected, and one new episode of hyperphosphatemia was observed. In conclusion, our study showed that the administration of a high-therapeutic dose of cholecalciferol allowed for a quick, effective, and safe leveling of 25(OH)D concentration in HD patients.

## 1. Introduction

Vitamin D is crucial for musculoskeletal and general health. Low levels are associated with an increased risk of death and morbidity, cancer, cardiovascular, autoimmune diseases, and infections. In hemodialysis (HD) patients it plays additionally an important role in the pathogenesis of Chronic Kidney Disease–Mineral And Bone Disorder (CKD-MBD) [1,2,3,4,5]. Serum 25-hydroxycholecalciferol (25(OH)D, calcidiol) concentration is the accepted marker for the evaluation of vitamin D status [6]. Vitamin D deficiency (25(OH)D < 20 ng/mL) and insufficiency (25(OH)D 20–29 ng/mL) are highly prevalent in CKD affecting over 80% of HD patients and requiring therapeutic intervention [6,7,8,9]. Despite the inadequate activation of vitamin D by hydroxylation in position 1-α in proximal tubules of the kidneys, the presence of extra-renal 1-α-hydroxylase results in the production of the active form of vitamin D, 1,25(OH)_2_D, calcitriol, in HD patients. That argues for the first-line use of cholecalciferol before the need for active vitamin D analogs [10].

Nephrological societies suggest the measurement of the level of 25(OH)D in CKD patients and the administration of cholecalciferol according to the guidelines for the general population [11]. However current guidelines for the prevention and treatment of vitamin D deficiency reveal a great heterogeneity of the recommendations, and in turn, indicate a special approach to CKD patients, so supplementation should be supervised and carried out cautiously [6,12]. So far, no cholecalciferol dosing regimen has been established in the clinical care of HD patients. 

The data from literature regarding the correction of vitamin D status in HD patients is miscellaneous and comes from studies with different protocols, dosing schedules, variable doses, and frequency of administration [10,13,14,15,16]. Some of the studies show that HD patients may require higher doses of cholecalciferol than the general population to achieve adequate serum levels of 25(OH)D. High doses appear to be safe and maybe more sufficient to increase 25(OH)D level > 30 ng/mL. However, effects on iPTH 1,25(OH)_2_D and other laboratory as well as clinical outcomes are ambiguous [14,15,16]. There are also significant knowledge gaps regarding the target level of 25(OH)D and the active metabolite of vitamin D, 1,25(OH)_2_D in HD patients, and the role of cholecalciferol in the therapy of CKD-MBD and its effect on overall outcomes. Therefore, our study aimed to evaluate the efficacy and safety of a high therapeutic dose of cholecalciferol in the correction of vitamin D3 insufficiency or deficiency in HD patients, following general recommendation, to reach the target serum 25(OH)D level > 30 ng/mL.

## 2. Materials and Methods

### 2.1. Study Design

This was an observational prospective study of a cohort of CKD stage 5, HD patients with vitamin D deficiency or insufficiency given oral cholecalciferol three times a week after each dialysis procedure from 4 January 2023 until the patient reached the target level of 25(OH)D > 30 ng/mL. Baseline serum 25(OH)D levels were assessed for all adult patients (*n* = 104) undergoing maintenance hemodialysis at our center in December 2022. The average vitamin 25(OH)D concentration was found to be 24.1 ± 14.95 ng/mL (median 19.15 ng/mL, range 5.3–83.3 ng/mL). A significant part of the patients, 77 out of 104 (74.04%) revealed 25(OH)D levels below 30 ng/mL. Based on exclusion criteria, 22 patients were selected for further analysis (Figure 1). The study protocol was approved by the Medical University of Gdańsk Ethical Committee (NKBBN/783/2022). All subjects provided their written informed consent to participate in the study. 

### 2.2. Treatment with Cholecalciferol

The patients were administered cholecalciferol (Devikap^®^, soft capsules, cholecalciferol 10,000 IU, 250 µg, POLPHARMA S.A., Starogard Gdański, Poland) at a dose of 70,000 IU/week (20,000 IU + 20,000 IU + 30,000 IU, immediately after each dialysis session). The observation was continued in the patient until the target level of 25(OH)D > 30 ng/mL was reached.

### 2.3. Laboratory Measurements

25(OH)D, 1,25(OH)_2_D, calcium, phosphate, and intact PTH (iPTH) concentrations were measured at 1–2-week intervals. Blood was drawn before a dialysis session, at baseline (T0), and in weeks T1, T2, T3, T4, T6, T7, T8, T9 of the treatment until 25(OH)D levels exceeded 30 ng/mL (Tmax). Five milliliters of blood were collected into a plain tube and allowed to clot at room temperature. The blood was then centrifuged and stored at minus 20 °C. Serum 1,25(OH)_2_D and 25(OH)D were measured using the chemiluminescent immunoassay method (LIAISON^®^ XL 1,25 Dihydroxyvitamin D, DiaSorin, Italy; laboratory normal range: 25–86.5 pg/mL, and LIAISON^®^ 25 OH Vitamin D TOTAL Assay (DiaSorin, Saluggia, Italy; optimal range for adults: 30–80 ng/mL). Based on the 95% Reference Interval, the following values were established for 1,25(OH)_2_D-median 47.8 pg/mL; observed range 2.5th to 97.5th percentile 19.9–79.3 pg/mL. iPTH (intact PTH) was measured by chemiluminescent assay (The DiaSorin LIAISON^®^ N-TACT^®^ PTH II Assay, DiaSorin, Saluggia, Italy; laboratory normal range: 11–67 pg/mL). Total serum calcium and phosphorus were measured by routine colorimetric method (Abbott GmbH, Wiesbaden, Germany). Serum calcium-phosphorus (CaxP) product was calculated. The evaluation of the safety profile involved monitoring episodes of hypercalcemia (defined as an elevation in calcium serum levels of more than 10.5 mg/dL) and episodes of hyperphosphatemia (defined as phosphorus serum levels of more than 4.6 mg/dL).

### 2.4. Statistical Analysis

Continuous data is reported as means (±standard deviation, SD) for normally distributed variables or medians (inter-quartile ranges, IQR) for non-normally distributed variables. The Shapiro-Wilk test was used to determine the distribution of continuous variables. Categorical data is reported as percentages of the total. The comparisons of results between the groups were performed using the nonparametric U-Mann–Whitney test. The Wilcoxon signed-rank test or ANOVA was used in the analysis comparing the results of the values of the study periods. Two-sided *p* < 0.05 was considered to be statistically significant. Lab results obtained from our patients periodically were used for the sample size calculation (*t*-test for dependent variables). A baseline 25(OH)D level of 16 ng/mL was predicted. We predicted an increase of 25(OH)D after our treatment to the recommended level of 30 ng/mL to give the study 80% power to detect such difference as statistically significant (*p* < 0.05, 2-tailed) with an expected standard deviation of 10 ng/mL, 10 patients had to complete the study. The statistical analysis was performed using the program Statistica 13.3 (TIBCO Software Inc.; Palo Alto, CA, USA).

## 3. Results

### 3.1. Characteristics of Patients

A total of 22 patients were included in the study, 16 men (73%) and 6 women (27%), with an average age of 72.5 ± 13.03 years. The 25(OH)D levels measured in the study group averaged 13.75 ± 4.83 ng/mL (a range of 6.9–23.9 ng/mL), 19 patients (86.36%) had levels below 20 ng/mL (deficiency), and 3 patients (13.64%) between 20 and 30 ng/mL (insufficiency). The dialysis protocols used were as follows: 18 patients underwent high-flux hemodialysis (HD), 2 patients received extended dialysis (HDx), 1 patient underwent online hemodiafiltration with predilution, and 1 patient underwent online hemodiafiltration with post-dilution: using a dialysate Ca concentration of 1.25 mmol/L (21 patients) or 1.5 mmol/L (1 patient). High-flux membranes (Helixone-Fresenius^®^, Bad Homburg vor der Höhe, Germany) or high-flux dialyzers with adequate selectivity for large proteins (Theranova, Baxter^®^, Deerfield, IL, USA), and ultrapure water were used. The median HD vintage at baseline was 38.9 months. During the follow-up, the renal replacement therapy protocol and doses of drugs affecting calcium and phosphate metabolism e.g., phosphate binders remained unchanged. Due to exclusion criteria, the patients did not receive active forms of vitamin D or calcimimetics. There were two patients with BMI > 30 kg/m^2^. A description of the study group is presented in Table 1.

### 3.2. 25(OH)D Levels

During the study, a statistically significant increase in 25(OH)D levels was observed among all patients (Table 2, Figure 2). All patients achieved the target value > 30 ng/mL, with a mean time of 2.86 ± 1.87 weeks. In the first week (T1), the target level of 25(OH)D was reached by 2 patients (9.09%), in T2, by 15 patients (68.18%), in T4, by 18 patients (81.18%), in T6, by 21 patients (95.46%), and, in T9, by all 22 patients (100%). In the study group were three patients with vitamin D insufficiency, two of them achieved the target 25(OH)D level after the first week, and the remaining patients achieved this level in the second week. The 90th percentile for the cumulative dose to achieve the target 25(OH)D concentration was 420,000 IU of cholecalciferol, while after normalization to body weight and BMI, it was 5675.7 IU/kg and 14,168.8 IU/[kg/m^2^], respectively. A presentation of the relationship between the number of patients reaching the 25(OH)D target concentration and the cumulative dose is shown in Figure 2. 

### 3.3. 1,25(OH)_2_D Levels

Cholecalciferol administration during the study resulted in the production of active vitamin D and a significant increase in 1,25(OH)_2_D levels was observed, (T0 vs. Tmax, 0.00 [0.00–6.38] ng/mL vs. 6.37 [8.11–15.60] ng/mL, *p* < 0.001, shown in Table 2 and Figure 3C). 

The median value increased during the observation period more than 6 times, however, only 2 patients (9.09%) achieved the lower limit of the reference range of 1,25(OH)_2_D, a concentration above 25 ng/mL. Moreover, a moderate correlation between 25(OH)D and 1,25(OH)_2_D concentrations was observed, with every 1 ng/mL increase in 25(OH)D corresponding to a 0.4 pg/mL increase in 1,25(OH)_2_D (r = 0.4102). This proportion of 1,25(OH)_2_D to 25(OH)D decreased with a higher 25(OH)D concentration level: for 25(OH)D concentrations below 30 ng/mL, the ratio was 0.60, and for levels above 30 ng/mL, it was 0.399. The correlation is shown in Figure 4. 

### 3.4. PTH Levels

The intact PTH concentrations did not change significantly during the observation period. Data is shown in Table 2 and Figure 3A.

### 3.5. Safety Assessment

No episodes of hypercalcemia were noticed throughout the observation period, and calcium levels did not increase. 14 of 22 patients (63.63%) had hyperphosphatemia before treatment inclusion. During follow-up, serum phosphorus concentrations were stable and only 1 new episode of hyperphosphatemia was observed. Patients had stable serum phosphorus concentrations, while the group with baseline hyperphosphatemia even experienced a decrease in phosphorus of 2.73% ± 13% on average. During follow-up, the number of patients with hyperphosphatemia decreased from 14 to 12 patients (giving a decrease of 14.3%). Table 2 and Figure 5 present data showing calcium and phosphorus concentrations.

## 4. Discussion

Our study confirms the high prevalence of vitamin D insufficiency and deficiency in patients with CKD undergoing HD therapy. In total, 74% of patients in our center showed 25(OH)D levels below 30 ng/mL. Irrespective of the factors related to age, gender, race, and obesity, some additional factors like impaired metabolism, prescribed dietary restrictions, and decreased sun exposure contribute to a low vitamin D status in CKD patients. Despite this unique situation, guidelines for HD patients are very imprecise, limited to suggesting correction of insufficiency or deficiency using treatment strategies recommended for the general population [11]. Therefore, even though the use of cholecalciferol in HD patients has been studied and implemented for years, variable doses and dosing schedules were used resulting in variable effects in correcting vitamin D status. 

The aim of this study was to check the effectiveness and safety of therapy with the maximum recommended therapeutic dose of cholecalciferol allowed in the general population. According to the Central and Eastern European Expert Consensus Statement published in 2022, a therapeutic dose of up to 10,000 IU of cholecalciferol daily may be used [6]. This dose also corresponds to the maximal physiologic dose rate per day (250 µg), which can be roughly acquired per day through exposure of whole-body skin to the sun for 30 min [17,18]. 

Cholecalciferol was administered to patients 3 times a week after each dialysis (20,000 IU + 20,000 IU + 30,000 IU of Devikap^®^). We have demonstrated an effective rapid increase of 25(OH)D level > 30 ng/mL in a mean time of 2.86 weeks. Over 80% of patients reached sufficient levels after 4 weeks of cholecalciferol input. However, there were patients requiring longer management up to 9 weeks. 25(OH)D levels significantly increased from 13.75 to 35.72 ng/mL. The literature data regarding doses’ timing and efficacy are very divergent. In many studies, the weekly dose was lower than in our study. Mieczkowski et al., proved that even small doses of cholecalciferol, 6000 IU/week, resulted in a significant rise in medians of 25(OH)D levels after 2 months from 11.3 to 37.5 ng/mL, with sufficient, constant levels in 100% HD patients [19]. This type of effectiveness has not been proven in other studies. Matuszkiewicz et al., showed a rise in 25(OH)D levels from 12.9 to 31.3 ng/mL using 12,000 IU/per week for 13 weeks and achieving levels > 30 ng/mL in 60% of HD patients [13]. Tokmak et al., used a dosing regimen with 20,000 IU cholecalciferol given weekly and mean 25(OH)D levels increased from 6.7 to 31.8 ng/mL at 26 weeks, however only 57% of HD patients became vitamin D sufficient [20]. Interestingly, Massart et al., obtained a similar effect using a dose of 25,000 IU weekly for only 13 weeks and reached normalization of 25(OH)D levels in 61%, even though the total dose of cholecalciferol was 2 times lower, most likely because the initial levels were higher (17.1 ng/mL at baseline, 35.2 ng/mL in final) [21]. However, Jean et al., with a higher total dose, of 100,000 IU of cholecalciferol given monthly, for 15 months reached levels > 30 ng/mL in 91% of patients [22]. There is also data on the effectiveness of higher doses and the more rapid correction of vitamin D status in HD patients. Wasse et al., conducted a study with cholecalciferol of 200,000 IU/week for 3 weeks, the mean 25(OH) levels increased from 13.3 to 52.4 ng/mL and 90.5% of subjects achieved serum 25(OH)D concentrations above 30 ng/mL [14]. Recently, Guella et al., were administered 300,000 IU monthly for 9 months. Serum levels of 25(OH)D increased significantly from 14.9 ng/mL to 39.6 ng/mL, and sufficient levels were obtained in 82.6% of the patients [16]. There are also regimens following the dose—response approach with the dose adjusted to 25(OH)D levels. Descombes et al., used an initial dose of 6000 IU per week for 2 months, and then an individualized dose (2000–12,000/week adjusted to the level). After 12 months, 86% of patients had 25(OH)D levels in the target range of 30–60 ng/mL [23]. Huish et al., used a 25(OH)D level adjusted dose of 40,000 IU per week followed by 20,000 fortnightly and successfully maintained serum levels between 30–60 ng/mL [24]. Matias et al., using the dose of 50,000 IU/per week or 10,000 IU per week depending on the degree of deficiency, and a maintenance dose of 2700 IU/3 × per week in vitamin D sufficiency, achieved 25(OH)D levels > 30 ng/mL in 94% patients at the end of 60-month study [25]. It seems that effectiveness may rely on the total dose, lower doses require a longer time to achieve a similar effect, and regimens that assume 25(OH)D monitoring and dose adjustment may be also effective.

Our regimen demonstrated correct vitamin D status with relatively low total doses. The 90th percentile for the total, cumulative dose to achieve the target 25(OH)D concentration was 420,000 IU of cholecalciferol, the total doses ranged from 70,000 IU to 630,000 IU. Similar effectiveness with a comparable proportion (at least 80%) of patients achieving levels > 30 ng/mL, was demonstrated in the studies with the total doses of 600,000 IU, 1,600,000 IU, 2,700,000 IU, as well as in studies with dose-to-level-adjusted regimens [14,16,22,23,24]. In any case, it is difficult to compare the total doses with the latter. Moreover, it is possible that the total dose to achieve correction of 25(OH)D depends on the initial levels. Armas et al., tried to quantify 25(OH)D response to cholecalciferol in HD patients and reported a rise of 0.56 ng/mL in 25(OH)D level per 40 IU cholecalciferol input per day, however, the findings did not seem to be easily adopted to clinical practice [26]. There are also reports on the use of doses adjusted for body weight, however low weekly doses of 100 IU/kg were used and only 27% of initially vitamin D-deficient dialysis patients achieved a level > 30 ng/m after 26 weeks [27]. Moreover, it is difficult to make conclusions about clinical effects in HD patients as patients on peritoneal dialysis and calcitriol therapy were included in this study. 

The intake of vitamin D may result in toxicity, which manifests with hypercalcemia, hypercalciuria, low PTH, hyperphosphatemia, and not necessarily increased 1,25(OH)_2_D concentration. Hypercalcemia does, however, usually not occur at 25(OH)D concentrations below 150 ng/mL, and until active vitamin D analogs are used. Already in the first studies on active vitamin D hypercalcemia and hyperphosphatemia, due to increased absorption induced by vitamin D receptor activators, in the gut were reported. Both high calcium and high calcium–phosphate product levels were identified as potential risk factors for the development of cardiovascular disease in CKD patients [28,29]. In opposition native vitamin D is unlikely to cause hypercalcemia, as 1-hydroxylase activation is regulated by PTH, FGF-23, 24–hydroxylase. However, the need for monitoring serum PTH, calcium, and phosphate levels is raised to prevent vitamin D toxicity in CKD [4]. In our study, regular monitoring of patients’ calcium, and phosphorus was done to assess safety. Our regimen was safe, without incidents of hypercalcemia, and the rise in phosphorus level, which is consistent with the literature data, even when the highest doses of native vitamin D were used [14,16]. Interestingly we observed a decrease in calcium levels and the number of patients with hyperphosphatemia decreased. A similar effect was shown in the study of Matias et al. [25].

The prevailing opinion is that PTH lowering with cholecalciferol in patients with CKD is unproven and clinically insignificant. As in most of the studies, we did not demonstrate the effect on PTH levels [13,19,20]. On the other hand, we might have prevented the increase in PTH levels, especially since the patients were not receiving any treatment other than phosphate binders at that time. Hypothetically higher doses of cholecalciferol and/or longer treatment time are necessary to decrease PTH. Jean et al., and Guella et al., demonstrated a decrease in PTH levels using pulse doses of 100,000 IU monthly and 300,000 IU monthly, respectively. The above effect was achieved after 1 and 9 months respectively [16,22]. On the other hand, Wasse et al., using pulse doses of 200,000 IU/per week for 3 weeks did not observe a significant decrease in PTH [14]. It also does not seem that the decrease in PTH depended on the achieved 25(OHD) levels. In some studies, a decrease in PTH was achieved with 25(OH)D levels reaching about 40 ng/mL [16,22,25], whereas the others did not demonstrate the effect on PTH even with 25(OH)D concentration of about 50 ng/mL [14,24]. There are also a number of studies showing improvement in PTH control with cholecalciferol, in which HD patients on active vitamin D analogues were not excluded [25,27,30,31]. Actually, the effect on PTH may also be dependent on the secondary hyperparathyroidism severity, and higher levels of 25(OH)D are required, or the PTH decrease depends on the 1,25(OH)_2_D correction.

According to the literature findings, we demonstrated a significant increase in active vitamin D—1,25(OH)_2_D levels when 25(OH)D levels were corrected. This confirms that correction of vitamin D deficiencies definitely contributes to the formation of active vitamin D in HD patients, and 1α-hydroxylase deficiency originating from renal tissue is not the clue problem in renal insufficiency, as was believed, since the presence of extra-renal 1-α-hydroxylase results in the production of the active form of vitamin D, 1,25(OH)_2_D, calcitriol, in HD patients. It is worth noting that the initial concentrations of 1,25(OH)_2_D were undetectable in many of our patients. Despite the increase (4.35 to 13.78 pg/mL), the mean levels remained under the laboratory norm (25–86.5 pg/mL), and in only 2 patients 1,25(OH)_2_D levels reached laboratory norms. Likewise, in most other studies, a significant rise in 1,25(OH)_2_D was reported, but the levels did not reach the normal values [13,16,19,22]. Perhaps the levels of 1,25(OH)_2_D, higher but still far below the normal range, explain the lack of episodes of hypercalcemia and hyperphosphatemia, and no effect on PTH in our study. In the current study, we have focused on the correction of vitamin D3 insufficiency or deficiency. Therefore, the observation in the patient was ended at the time, when the level > 30 ng/mL was recorded. It seems that it may be interesting to continue therapy to reach the upper normal ranges for 25(OH)D and demonstrate the effect on 1,25(OH)_2_D levels and PTH. This will be the subject of our further research.

The main limitations of our study are the lack of a control group treated with a placebo, a relatively small number of patients not including diabetics and obese patients, and short-term observation. The latter should be borne in mind, given the fact vitamin D, being fat-soluble, can accumulate and maintain an action in time. The presented study is observational and exploratory in nature and is intended to be a contribution to further research. To identify benefits, risks, and adverse side effects, a larger and longer study is warranted.

The study’s strength was the administration of cholecalciferol under supervision at the dialysis unit which ensured good compliance. However, a study comparing different dosage regimens, e.g., intermittent higher vs. daily dosing regimens would be interesting. Another unique strength of the study was that the patients did not take the active form of vitamin D or vitamin D analogues many months before and during the study. Thus, the observed effect could be attributed solely to the cholecalciferol administration. Finally, our study provides real-life data reflecting a relatively diverse cohort of an average dialysis unit. The fact that all subjects achieved the recommended 25(OH)D target, makes our observation readily applicable to any other population of HD patients.

## 5. Conclusions

In conclusion, our study showed that the administration of a high therapeutic dose of cholecalciferol under supervision at the dialysis unit allowed for a quick, effective, and safe leveling of 25(OH)D concentration in HD patients. The significant rise in 1,25(OH)_2_D and no impact on PTH levels was also shown. Further studies are necessary to determine the safety of high-dose cholecalciferol in long-term use, as well as target levels of 25(OH)D with satisfactory impact on 1,25(OH)_2_D and PTH levels.

## Figures and Tables

**Figure 1 biomedicines-12-00377-f001:**
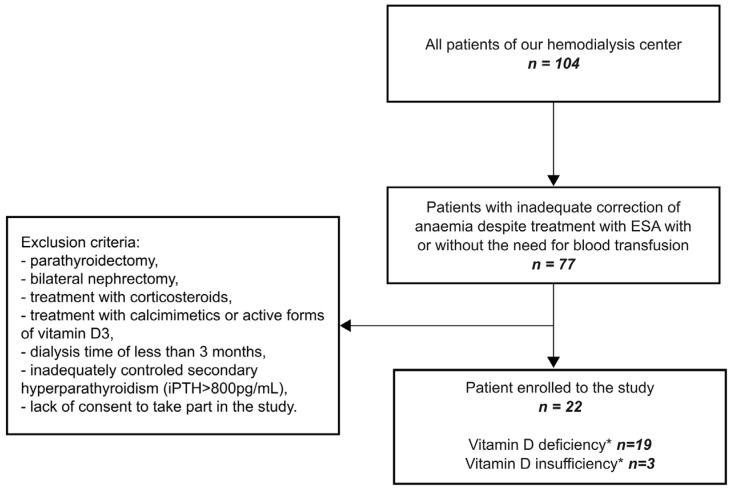
The flow diagram of patient selection. * = vitamin D deficiency 25(OH)D < 20 ng/mL; vitamin D insufficiency 25(OH)D 20–29 ng/mL).

**Figure 2 biomedicines-12-00377-f002:**
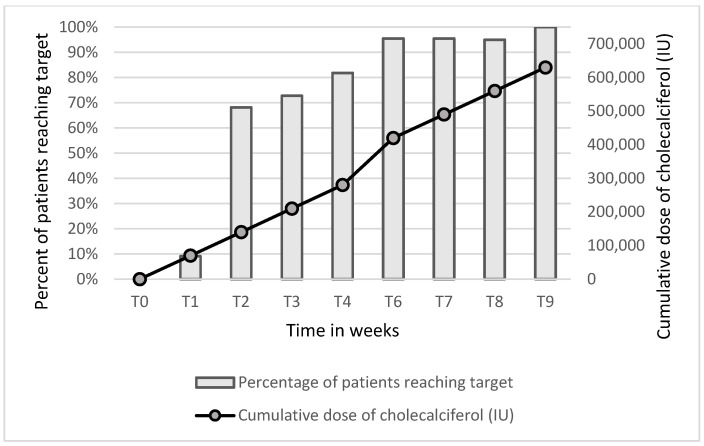
Number of patients who reached the target 25(OH)D concentration (bars) and the cumulative dose of cholecalciferol (line).

**Figure 3 biomedicines-12-00377-f003:**
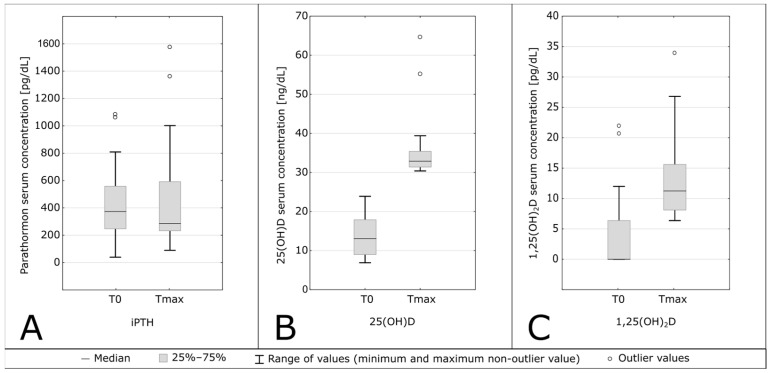
Serum concentrations of intact parathormone (iPTH, part (**A**)), calcidiol (25(OH)D, part (**B**)), and calcitriol (1,25(OH)_2_D, part (**C**)) before supplementation (T0) and after reaching the target serum 25(OH)D concentration (Tmax).

**Figure 4 biomedicines-12-00377-f004:**
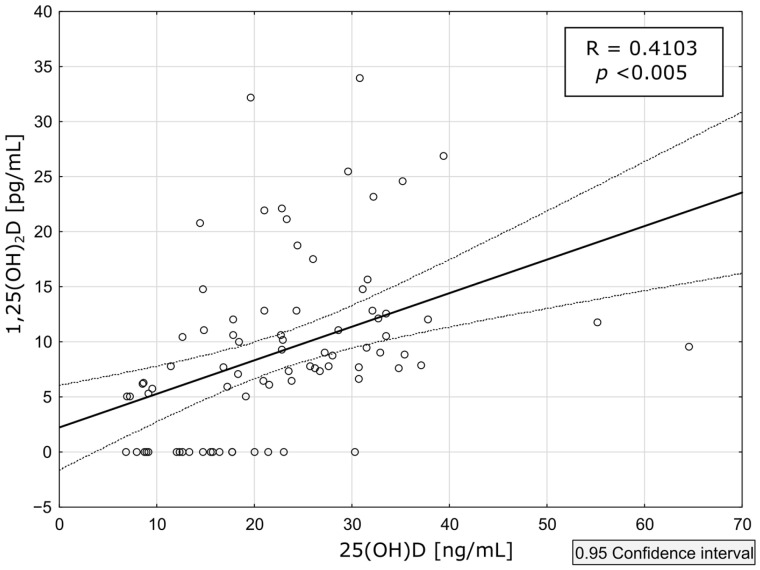
Correlation between serum concentrations of calcidiol (25(OH)D), and calcitriol (1,25(OH)_2_D) during the therapy with cholecalciferol. Each white circle represents each data point, both types of lines represent the 0.95 confidence interval value as described above.

**Figure 5 biomedicines-12-00377-f005:**
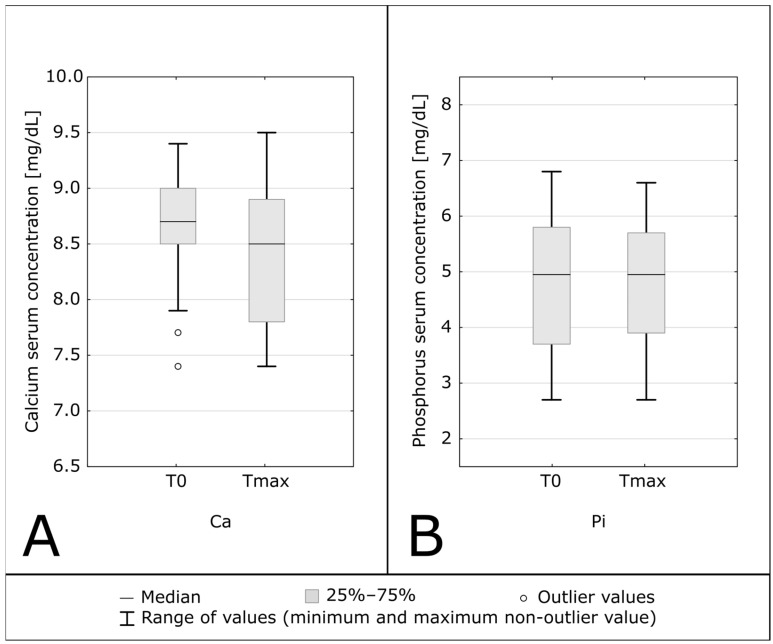
Serum concentrations of calcium (Ca, part (**A**)), and phosphorus (Pi, part (**B**)) before supplementation (T0) and after reaching the target serum calcidiol concentration (Tmax).

**Table 1 biomedicines-12-00377-t001:** Characteristics of the study group.

Gender (Men/Women)	16/6
Causes of ESRD (n/%)	
Ischemic nephropathy/no histology	5/22.7
Autosomal dominant polycystic kidney disease	4/18.2
Diabetic nephropathy in type 2 diabetes/no histology	6/27.7
Glomerulonephritis/no histology	2/9.1
Chronic hypertensive nephropathy/no histology	1/4.5
Congenital renal malformation	1/4.5
Unknown	2/9.1
Age (years)	72.5 (67–81)
Charlson Comorbidity Index (points)	8 (2–14)
Dialysis vintage (months)	25.0 (12–56)
Body Mass Index (kg/m^2^)	25.27 (22.50–28.76)

**Table 2 biomedicines-12-00377-t002:** Comparison of parameters at the starting point of the study (T0) and at the point of achieving the target 25(OH)D concentration (Tmax).

Parameters	T0	Tmax	*p*
25(OH)D [ng/mL]	13.05 (9.00–17.90)	32.9 (31.4–35.4)	<0.001
1,25(OH)_2_D [pg/mL]	0.00 (0.00–6.38)	6.37 (8.11–15.60)	<0.001
iPTH [pg/mL]	373.50 (247–558)	236 (146–537.5)	0.70
Calcium [mg/dL]	8.7 (8.5–9.0)	8.5 (7.8–8.9)	0.048
Phosphorus [mg/dL]	4.95 (3.7–5.8)	4.95 (3.9–5.7)	0.68
Serum CaxP [mg^2^/dL^2^]	48.778 (42.416–52.7)	43.7 (36.16–48.36)	0.046

T0, before supplementation; Tmax, at the time when 25(OH)D concentration reached a level above 30 ng/mL.

## Data Availability

The data are available from the corresponding authors upon reasonable request.

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
