# Peer review of "The Efficacy and Safety of High-Dose Cholecalciferol Therapy in Hemodialysis Patients"

_biomedicines, 2024, doi:10.3390/biomedicines12020377_

Round 1

Reviewer 1 Report

Comments and Suggestions for Authors

The safety and efficacy of nutritional vitamin D in dialysis patients with SHPT is an area of controversy. The reviewer believes that additional research is needed in this field. However, this study has several limitations.

1.      The sample size is relatively small.

2.      The study did not include a comparison group treated with placebo. The study is a single-arm, interventional study and not a randomizec controlled trial.

3.      The results are confirmatory in nature.

4.      The study did not assess additional outcomes of interest, such as effects of nutritional vitamin D on arterial structure (PWV, other novel biomarkers implicated in the clacification process).

5.      Background treatment with active vitamin D was not reported.

6.      In fact, there is a need for a future RCT to provide a direct comparison between nutritional vitamin D and active vitamin D analogs. This needs to be stated in the discussion. Some prior studies conducted in patients with predialysis CKD need to be discussed.

7.      The authors should provide a sample size calculation in the methods.

Author Response

The safety and efficacy of nutritional vitamin D in dialysis patients with SHPT is an area of controversy. The reviewer believes that additional research is needed in this field. However, this study has several limitations

We agree that this is a small, preliminary, exploratory study with several limitations (given in the discussion), line 361. The study is observational and descriptive in nature. The same subjects were subjected to multiple tests and served as their own control group. The study was aimed at a preliminary assessment of the effectiveness and safety of the therapy used, which was allowed by the research methodology used. We decided to conduct this study because there is little research analyzing the effects of high doses of cholecalciferol in the population of hemodialysis patients and there are lack of appropriate treatment recommendations in such a population (novelty). Moreover, some clinicians question the use of this treatment due to the potential lack of activation of the vitamin D in the kidneys in chronically dialysis patients. The strength and novelty of this analysis was also that 100% treatment compliance was ensured (controlled drug administration). Of course, we are aware that the results of our exploratory study are preliminary and can only be an argument for conducting controlled studies. This is what science is all about, where observational research precedes subsequent study stages.

1. The sample size is relatively small.

We agree with Reviewer’s comment. The sample size is relatively small and this is one of our’s study limitations – the detailed information about limitations was included in the discussion, line 361. The small size of the study group resulted of strict inclusion /exclusion criteria (shown in Figure 1. The flow diagram of patients selection), e.g. PTH values and no treatment with an active form of vitamin D and/or calcimimetics. On the other hand, for an exploratory assessment of the significance of the large changes in 25(OH) levels that we expected, the group size we used is completely sufficient, as indicated by the analysis of the necessary sample size carried out in accordance with the reviewer's suggestion (Methods, line 127).

  1. The study did not include a comparison group treated with placebo. The study is a single-arm, interventional study and not a randomized controlled trial.

The study was designed as an observational prospective study of a selected group of HD patients with vitamin D deficiency or insufficiency given nutritional vitamin D (oral cholecalciferol). The aim was to evaluate the efficacy and safety of high therapeutic dose of cholecalciferol in correction of vitamin D status, following general recommendation (therapeutic dose of 10,000 jm/day and target serum 25(OH)D level > 30 ng/mL). The same subjects were subjected to multiple tests and served as their own control group. As it was an observational study, we did not use a placebo which we clearly point out in the discussion.

  1.  The results are confirmatory in nature.

We agree with the Reviewer, that our study consent with some data known from the literature (and we discuss that in Discussion) such as high prevalence of vitamin D insufficiency and deficiency in HD patients (Line 223), and significant increase in 25(OH)D (line 244) and 1,25(OH)2D (line 345) concentrations in HD patients treated with cholecalciferol. However, the unique value of the study was the evaluation of high therapeutic doses of vitamin D in obtaining the levels recommended for the general population. We used the recommended therapeutic doses recommended in general population and adjusted it to the dialysis regimen (line 235). Consensus practically does not apply to hemodialysis patients. There are only recommendations, that screening of vitamin D deficiency should be considered in the some patients/individuals or conditions: a.o.chronic kidney disease, and considerations also concern the use of calcidiol and active vitamin D in chronic kidney diseases. So, starting with general recommendations, we want to contribute in the discussion on specific recommendations for HD patients. Additional study's strengths are given in line 374.

  1. The study did not assess additional outcomes of interest, such as effects of nutritional vitamin D on arterial structure (PWV, other novel biomarkers implicated in the calcification process).

In the current study we have focused on the correction vitamin D3 status. However, this is an initial study. We continue further research, especially in the field of secondary hyperparathyreoidism, and some additional outcomes (e.g. calcification biomarkers) will be included. We discuss impact on PTH in Discussion (Line 325).

  1. Background treatment with active vitamin D was not reported.

Due to exclusion criteria the patients were not on active vitamin D and/or cinacalcet, the information is included in Figure 1 and was added in Characteristics of patients (line 154).

  1. In fact, there is a need for a future RCT to provide a direct comparison between nutritional vitamin D and active vitamin D analogs. This needs to be stated in the discussion. Some prior studies conducted in patients with predialysis CKD need to be discussed.

The aim of the study was the assessment of vitamin D status correction with high therapeutic doses of nutritional vitamin D, and active vitamin D analogs are not recommended for this purpose. However, we agree that further research is needed. Our study showed that correcting 25(OH)D > 30 ng/mL had no impact on PTH. We continue our research and speculate that higher levels of 25(OH)D might demonstrate results in higher 1,25(OH)2D levels and suppression in PTH. The information was included in the discussion. (line 362).

  1. The authors should provide a sample size calculation in the methods.

It was done as requested as follows (Methods, Line 127):

Lab results obtained from our patients periodically were used for the sample size calculation (t-test for dependent variables ). Baseline 25(OH)D level of 16 ng/mL was predicted. We predicted an increase of 25(OH)D after our treatment to recommended level of 30 ng/mlL To give the study 80% power to detect such difference as statistically significant (P<0.05, 2-tailed) with an expected standard deviation of 10 ng/mL, 10 patients had to complete the study.

Reviewer 2 Report

Comments and Suggestions for Authors

To the authors,

Your prospective study is a well-done, methodologically strong foray into vitamin D supplementation in hemodialysis patients. Besides the results that you obtained and presented, which are hypothesis-generating and a strong foundation for further research, you have also provided a succint overview of the heterogeneity in the evidence on vitamin D supplementation in HD patients and correctly pointed out to the lack of specific dosing regiments and recommendations in the nephrology societies' guidelines. I am opined that this study may be a good start and an important piece in solving the puzzle of vitamin D supplementation dosing in HD patients. 

Comments on the Quality of English Language

Some minor edits are required, please proofread the text several times and correct the minor errors in grammar and synthax that you detect. 

Author Response

To the authors,

Your prospective study is a well-done, methodologically strong foray into vitamin D supplementation in hemodialysis patients. Besides the results that you obtained and presented, which are hypothesis-generating and a strong foundation for further research, you have also provided a succint overview of the heterogeneity in the evidence on vitamin D supplementation in HD patients and correctly pointed out to the lack of specific dosing regiments and recommendations in the nephrology societies' guidelines. I am opined that this study may be a good start and an important piece in solving the puzzle of vitamin D supplementation dosing in HD patients.

Thank you for your review, this is exactly what we wanted to achieve in our study, we couldn't summarize it better. This is a preliminary , observational and exploratory study and was intended to be a contribution to further research.

Comments on the Quality of English Language

Some minor edits are required, please proofread the text several times and correct the minor errors in grammar and synthax that you detect.

Thank you for your comments. We will carry out careful proofreading

Round 2

Reviewer 1 Report

Comments and Suggestions for Authors

The reviewers have provided satisfactory answers and revisions. This is a pilot study that provides the rationale for further research in this area.